# Hedgehog Signaling in Cortical Development

**DOI:** 10.3390/cells13010021

**Published:** 2023-12-21

**Authors:** Eva Cai, Maximiliano Gonzalez Barba, Xuecai Ge

**Affiliations:** Department of Molecular and Cell Biology, School of Natural Sciences, University of California Merced, Merced, CA 95340, USA

**Keywords:** Hedgehog signaling, neocortex, cortical development, knockout mouse model, brain gyrification, radial glial cells, neural patterning, neurogenesis

## Abstract

The Hedgehog (Hh) pathway plays a crucial role in embryonic development, acting both as a morphogenic signal that organizes tissue formation and a potent mitogenic signal driving cell proliferation. Dysregulated Hh signaling leads to various developmental defects in the brain. This article aims to review the roles of Hh signaling in the development of the neocortex in the mammalian brain, focusing on its regulation of neural progenitor proliferation and neuronal production. The review will summarize studies on genetic mouse models that have targeted different components of the Hh pathway, such as the ligand Shh, the receptor Ptch1, the GPCR-like transducer Smo, the intracellular transducer Sufu, and the three Gli transcription factors. As key insights into the Hh signaling transduction mechanism were obtained from mouse models displaying neural tube defects, this review will also cover some studies on Hh signaling in neural tube development. The results from these genetic mouse models suggest an intriguing hypothesis that elevated Hh signaling may play a role in the gyrification of the brain in certain species. Additionally, the distinctive production of GABAergic interneurons in the dorsal cortex in the human brain may also be linked to the extension of Hh signaling from the ventral to the dorsal brain region. Overall, these results suggest key roles of Hh signaling as both a morphogenic and mitogenic signal during the forebrain development and imply the potential involvement of Hh signaling in the evolutionary expansion of the neocortex.

## 1. Introduction

### 1.1. Transduction of Hh Signaling

The Hh pathway is unique for its distinctive mode of transduction, which hinges on a series of protein de-repression, rather than activation, to facilitate signaling (Figure 1). In the mammalian system, at the resting stage, the 12-transmembrane protein Patched (Ptch) resides in the primary cilium and inhibits the G-protein coupled receptor (GPCR) family protein Smoothened (Smo) [1,2]. While Smo is inactive, two inhibitory proteins, the Suppressor of fused (Sufu) and Protein Kinase A (PKA) suppress the activity of Gli transcription factors. Sufu binds to Gli, preventing its translocation into the nucleus [3] and promoting its phosphorylation by PKA [4]. Phosphorylated Gli proteins are proteolytically processed to repressor forms [3,5,6]. In the presence of the Sonic Hedgehog (Shh) ligand, Shh binds to the receptor Ptch, followed by the exit of the Ptch-Shh complex from the cilium. This in turn allows Smo accumulation and activation in the cilium. Through the PKI motif in its cytoplasmic tail, active Smo in the cilium binds to and inhibits PKA catalytic subunit [7]. This subsequently enables the activation of Gli transcription factors, which then translocate to the nucleus to initiate the transcription of downstream target genes [3,8]. For more in-depth information regarding the biochemical and molecular mechanisms of Hh signal transduction, we refer the readers to other excellent comprehensive reviews on this topic [9,10,11]. The non-canonical Hh signaling refers to cellular responses that are independent of Gli transcription factors and is not the focus of this review.

### 1.2. A Brief Overview of Corticogenesis

In mice, the closure of the neural plate starts at embryonic day 8.5 (E8.5), initiating at the hindbrain and progressing bi-directionally toward the forebrain and the spine. By E10.5 a fully closed neural tube is formed [12]. Neuroepithelial cells in the neural plate divide symmetrically to populate the developing neural tube. Later, most neuroepithelial cells transit into radial glial cells (RGCs) after acquiring some of the astroglia characteristics [13,14]. RGCs first undergo self-renewing symmetric division to expand the neural progenitor pool, followed by asymmetric division to produce neurons [15,16,17,18,19]. The newly differentiated neurons migrate to the cortex along the radial fiber of RGCs [20,21]. In the middle stage of cortical neurogenesis, RGCs also produce intermediate neural progenitors (INPs) that undergo one or two cell divisions in the subventricular zone (SVZ) to generate more neurons. This indirect neurogenesis via INPs is a characteristic feature of mammals and is believed to promote expansion of the cerebral cortex and its excitatory neuronal population [22,23] (Figure 2).

Neurons populate the developing cortex in an inside–out manner; that is, later-born neurons migrate past the early-born neurons to settle down at the more superficial cortical layers. As a result, the excitatory neurons in the cortex are organized into six layers roughly based on their birthdate. Early born neurons occupy the deep layers, whereas later-born neurons reside in the superficial layer [24]. The disruption of layer specification was observed in several human genetic diseases, such as doublecortin and lissencephaly [25,26]. The impaired neuronal migration is partially responsible for the brain structural deficits in these patients.

A type of highly neurogenic progenitors, the basal radial glial cells (bRGCs), were found to be abundant in the gyrencephalic species, such as the human brain (Figure 2). bRGCS are thought to be important for the dramatic expansion of cortical neuronal population and the gyrification (the folds at the brain surface) of the human brain [27,28,29]. In lissencephalic species, such as rodents, these cells are much fewer and has only limited neurogenic potential [30]. It remains an intriguing question what genomic features are responsible for the expansion of this type of neural progenitors during evolution. For thorough overviews of human cortical development, we refer readers to existing reviews dedicated to this specific topic [31,32,33,34].

In neurodevelopment, Hh signaling plays essential roles in cerebellar development [35,36,37]. Furthermore, non-canonical Hh signaling has also been identified to be crucial in axon guidance [38,39,40]. However, our present review will not emphasize these brain regions. Instead, we will focus on roles of Hh signaling in the neural progenitor proliferation and neural production during neocortical development. Within this scope, we will summarize results from genetic mouse models targeting core components of the Hh pathway. In certain instances, we will also highlight studies that delineate the interaction of Hh signaling components in telencephalon patterning, as findings from these studies have contributed to key mechanistic insights into Hh signal transduction.

## 2. The Hypothesis of Expanded Hh Signaling in Gyrification of the Brain

Gyrencephalic species have folded cortices with gyri and sulci, enabling a greater surface area relative to cranial size. Gyrification is believed to associate with higher cognitive function and intelligence [41]. To produce a gyrified brain, progenitors in the forebrain must undergo substantially more divisions than these in lissencephalic brains. The expanded proliferation of bRGCs in the outer subventricular zone has been shown to contribute to increased neurogenesis in primates and other gyrencephalic species [42,43] (Figure 2). A few studies reported that in rodents, a lissencephalic species, over-activation of Hh signaling promoted RGC proliferation and increased bRGC specification, which eventually leads to gyrification-like folds at the surface of the rodent brain [44,45,46]. Therefore, it is postulated that Hh signaling may be more prominently active in gyrencephalic brains, potentially contributing to the enlargement and gyrification of the cerebral cortex [47].

A few studies show that over-activating Hh signaling in the developing mouse cortex results in an increase in the upper neuron population and the formation of cortical folds. In one study, Hh signaling is overactivated in RGCs by the *SmoM2* transgene driven by *GFAP-Cre*. SmoM2 is a constitutive active mutant of Smo that triggers Hh signaling, independently of Shh. In this transgenic mouse model, both the bRGC and INP populations are increased in the neocortex [44]. The induced bRGCs displayed bipolar morphology closely resembling that of bRGCs in gyrencephalic species. The transgenic mouse brain exhibited higher cell density in the cortex, along with the development of folds in the upper layers of the cingulate cortex. Moreover, the formation of cortical folds relies on the primary cilia and Gli2, as genetically eliminating cilia or depleting Gli2 disrupts this phenotype. The bRGCs and INPs in *SmoM2-flox*;*GFAP-Cre* mice exhibited increased proliferation and self-renewal. Conversely, deleting *Smo* in RGCs via the same *GFAP-Cre* results in a reduced number of INP and fewer bRGCs, ultimately leading to a reduction in brain size [44]. In another study, both Hh signaling and Notch signaling were activated in a transgenic mouse model [45]. The activated Hh signaling, mediated through the *SmoM2* transgene, aims to promote the proliferation of neural progenitors; and the activated Notch signaling, mediated through the *Hes1* transgene, maintains an undifferentiated state in the neural progenitors. The dual activation of Hh and Notch signaling led to a dramatic increase in pax6-positive neural progenitors and remarkable enlargement of the neocortical surface area. Furthermore, obvious folds were observed at the neocortical surface [45].

The critical roles of Hh signaling in cortical gyrification are demonstrated by studies in the naturally gyrencephalic ferret brain. One study found that a subset of bRGCs expressing *Hopx* gene plays important roles in the gyrification of the ferret cortex. The number and self-renewal proliferation of Hopx-positive bRGCs rely on Hh signaling. Suppressing Hh signaling reduced the number of Hopx-positive bRGCs and cortical folding, while enhancing Hh signaling with in utero electroplated Shh-N has opposite effects [47]. Aligned with these results, activating Hh signaling with the Smo agonist, SAG, increased the self-renewal of bRGCs and INP proliferation in the ferret developing cortex, whereas suppressing Hh signaling with Smo antagonist, vismodegib, has opposite effects. Notably, at early developmental stage, SAG also increased the bGRC-producing division modes in the apical RGCs [46]. Together, these findings suggest that Hh signaling could serve as a key regulator of mammalian brain evolution.

## 3. Roles of Hedgehog Signaling during Cortical Development

The mitogenic roles of Hh signaling is best described in the developing cerebellum where Shh is secreted from Purkinje neurons to stimulate the proliferation of granule neuron progenitors (GNPs) [35,36,37]. In the developing forebrain, the relationship between Hh signaling and cell proliferation is more complex. Here, we will review how Hh signaling regulates neurogenesis during cortical development in the forebrain. We will summarize studies that involve the targeted knockout of different components of the Hh pathway at various developmental stages to the elucidate roles of Hh signaling during cortical development. As numerous key insights into the Hh signaling transduction mechanisms were derived from mouse models exhibiting neural tube development defects, we will also briefly summarize the phenotypes observed in these mouse models. Table 1 compiles information on the mouse mutants referenced in this article.

### 3.1. Roles of Shh, the Ligand of Hh Pathway, in Cortical Development

In humans, the loss of function of Hh signaling leads to holoprosencephaly, a severe brain malformation characterized by incomplete separation of the forebrain during early embryonic development [48,49,50]. In mouse models, global knockout of *Shh* leads to severe disruption of the patterning in the developing central nervous system [51,52]. Shh-null mice die at birth and exhibit striking holoprosencephaly. The head structure was reduced to a proboscis-like extension with a single-fused telencephalon. The brain showed no ventral telencephalic structures, and normally dorsal telencephalic genes such as Pax3 and Pax6 invade the ventral region [51]. To circumvent the essential roles of Shh in tissue patterning during early embryonic development, conditional *Shh*-knockout mice were generated with *Emx1-Cre* that mediates genetic recombination in forebrain excitatory neurons [53]. At early developmental stages (E13.5), BrdU incorporation results suggested reduced cell proliferation and reduced neuronal numbers in the superficial layers. Further analysis found that neural progenitor cells in mutant mice exhibited prolonged cell cycles and slowed dynamics to exit and re-enter the cell cycle. Interestingly, at a later developmental stage (E15.5), the cell cycling defects were recovered. This recovery can probably be attributed to the arrival of Shh from other sources, such as the migrating interneurons from ventral regions. Overall, the conditional knockout of *Shh* by *Emx-Cre* leads to the diminished size of the dorsal telencephalon [51]. These results suggest that Hh signaling is important in maintaining the neural progenitor pool in the developing forebrain.

While the loss of Shh leads to the loss of ventral structures and decreased cortical size, the gain of function of Shh promotes ventral identities and the growth of dorsal forebrain. In a study using mouse neocortical explants, Shh treatment led to increased BrdU incorporation in postnatal day 3 (P3) explants, whereas the anti-Shh antibody led to reduced proliferation of neural progenitors in the VZ/SVZ. In this region, Gli1 was also appropriately upregulated or downregulated [54]. Along the same line, embryos overexpressing Shh via in utero electroporation at E12.5 exhibited expanded cortical plate (CP) and SVZ/VZ in the dorsal cortex. The cortex of the electroporated hemisphere was almost double the width of the control hemisphere. Tbr2-positive intermediate neural progenitors in the SVZ were also increased [53]. In situ hybridization results showed that all three Gli transcription factors are expressed in the proliferative zone VZ/SVZ at the peak of neurogenesis (E15.5-E17.5) [54]. These results suggest that Shh acts as a mitogen for the neural progenitors in the dorsal telencephalon.

### 3.2. The Receptor Ptch1 and Its Coreceptors in Cortical Development

In human patients, gain-of-function mutations in Ptch1 gene, through partial or whole gene duplication, lead to microcephaly [55,56], a brain developmental defect characterized by small head size and the thinning of the cerebral cortex. The reduced brain size is attributed to the attenuated Hh signaling due to overactive Ptch1. Conversely, loss-of-function mutations of *Ptch1* lead to holoprosencephaly, a characteristic phenotype of reduced Hh signaling [57]. In mammals, the two Shh receptors, Ptch1 and Ptch2, share 56% homology [58]. However, Ptch1 mediates most of the Hh signaling related developmental events, as *Ptch2*-null mouse are grossly normal [59]. We will hence focus on studies on Ptch1. Consistent with its role as a suppressor of Hh signaling, *Ptch1* duplication in the human genome is associated with microcephaly and developmental delay [59]. In the developing neural tube, Ptch1 is expressed in a gradient from ventral to dorsal [60,61]. Mice with homozygous knockout of *Ptch1* do not survive past embryonic days 9–10.5 (E9-10.5) and exhibit open and overgrown neural tubes, which is a typical hallmark of excessive Hh signaling activity. *Ptch1* heterozygous mutants are viable but exhibit a high tendency to developing medulloblastoma in the cerebellum [61]. When *Ptch1* is overexpressed in the developing cortex driven by the *Nestin* enhancer (expression starting at E9), the mutant embryos exhibit lower expression levels of Hh target genes, accompanied by dorsalization of the neural tubes—cells of dorsal identity invade into the ventral region [60].

To specifically eliminate Ptch1 from the neocortex, Allen BL and colleagues generated a Ptch1 conditional knockout mouse model mediated by *Nestin-Cre* [62]. The elevated Hh signaling in the mutant embryos dramatically expanded the number of neural progenitor cells. The cortex appeared irregular in its thickness, with lamination defects due to improper neuronal production. Elevated Shh signaling significantly shortened the duration of the cell cycle. Cultured RGCs from E14.5 brains of the *Nestin-Cre*;*Ptch1-flox* mice display increased self-renewing symmetric cell divisions and decreased neurogenic divisions [62]. In summary, *Ptch1* gain-of-function reduces Hh signaling in the developing cortex, decreases cortical size, and leads to developmental defects such as microcephaly. Loss-of-function mutations in *Ptch1* elevate Hh signaling and increase the thickness of the cortex. This is accompanied by an expanded pool of NPCs, suggesting the pivotal role of Hh signaling in NPC proliferation and the regulation of cortical size.

Several membrane-associated Shh-binding proteins, including CAM-related/downregulated by oncogenes (Cdon), brother of CDO (Boc), and Growth arrest-specific 1 (Gas1), function as Ptch1 coreceptors. Most studies so far reveal that these coreceptors play essential roles in the patterning of the central nervous system. *Cdon*-null mice exhibit holoprosencephaly [63,64], a characteristic phenotype indicative of severe reduction in Hh signaling. The proper function of Cdon requires its interaction with both Shh and Ptch1, as mutations disrupting Cdon-Ptch1 interaction results in holoprosencephaly [65]. *Boc*-null embryos display normal neural patterning, and Gas1-null mice show mild holoprosencephaly [66,67]. However, *Cdon/Boc* and *Cdon/Gas1* double mutants exhibit more severe forms of holoprosencephaly than any single mutant [66,67,68]. Together, these results suggest a complicated network of Hh signaling regulation at the cell surface. Despite these insights into neural patterning, the involvement of these Ptch1 coreceptors in the regulation of neocortical formation remains unexplored. While some coreceptors have been reported to be expressed in RGCs [69], there is a paucity of studies in this direction. Addressing this question requires the use of conditional knockout mouse models to circumvent the early disruption of neural patterning seen in global knockout mice.

### 3.3. The GPCR-like Signaling Transducer, Smoothened, in Cortical Development

Smo sits at the nodal position in Hh signal transduction. When *Smo* is genetically ablated in mice, it results in the complete abolishment of Hh signaling, leading to severe neurodevelopmental defects. Interestingly, *Smo*-knockout mice exhibit more pronounced neurodevelopmental defects compared to *Shh*-knockout mice. This milder severity in *Shh*-knockout mice is likely attributed to the compensatory effects of other *Shh* paralogs, namely *Ihh* and *Dhh*. *Smo* knockout circumvents this redundancy of ligands [53,70].

In early brain development, Smo is important for patterning the ventral telencephalon. Conditional *Smo* knockout mediated by *FoxG1-Cre* eliminated Smo by E9 [71]. At E10.5, *Smo* conditional-knockout (cKO) mice had distinguishably smaller telencephalons compared to wild-type (WT) littermates. In *Smo* cKO mice, the ventral telencephalon failed to develop, as evidenced by the absence of ventral markers such as Nkx2.1 and Gsh2. Conversely, the dorsal marker Pax6 expanded across the entire neuroepithelium. As a result, cell types originated from the ventral region, such as interneurons and oligodendrocytes, are nearly absent from *Smo* cKO mice [67].

Following the establishment of the dorsal–ventral pattern, Smo plays a critical role in regulating the size of the telencephalon and affecting the proliferation and survival of progenitor cells in the dorsal cortex. In an elegant study with two mouse models, *Smo* conditional knockout was induced by either *Emx1-Cre* (initiated at E10.5) or *Nestin-Cre* (initiated at E12.5 KO). No significant telencephalon patterning abnormalities are observed in the brain of either mice [53,72]. However, in *Emx1-Cre* driven *Smo* cKO mice, the telencephalon is smaller than their WT littermates. Examination of coronal and parasagittal sections revealed a reduction in the size of the dorsal telencephalon, whereas the morphology of the ventral telencephalon is not affected. In addition, BrdU incorporation assay suggested reduced cell proliferation in the mutant mouse brain. Further analysis revealed that the proliferative defects in the mutants were due to a prolonged cell cycle and reduced cell cycle exit [53]

In mice of *Smo* cKO driven by *Nestin-Cre*, the patterning of telencephalon appeared largely normal [72]. The only obvious patterning abnormalities were the decreased size of the MGE and the reduced production of early oligodendrocyte precursors. Instead, mutant mice show defects in the brain regions where neurogenesis continues in the adult, such as the dentate gyrus and the olfactory bulb. In these brain regions, there was a decrease in number of proliferating cells at the perinatal stage. Further, progenitors from the SVZ of the mutant mice exhibited reduced potential to generate neurospheres. These results suggest that Hh signaling is important in maintaining the neural stem cell niche in the perinatal telencephalon [72].

### 3.4. The Negative Hh Regulator, Sufu, in Cortical Development

The Suppressor of the Fused (Sufu) serves as a negative regulator of the Hh pathway. At the signaling resting stage, Sufu binds to the Gli transcription factors, and the complexes translocate through the cilia. Subsequently, the Gli proteins are processed into repressor forms. These Gli repressors then enter the nucleus and inhibit the transcription of Hh-target genes. When Shh is present to activate the Hh pathway, Sufu-Gli complexes travel through the cilium, resulting in the activation of Gli transcription factors [3,8]. Nevertheless, the precise mechanisms by which Sufu-Gli cilia trafficking influences Gli protein processing remain largely unknown.

Complete loss of *Sufu* from embryos resulted in embryonic lethality by E9.5. Examination of the embryo revealed severely deformed cephalic region, with open neural tube and fore-, mid-, and hindbrain [73]. Further examination unveiled a full ventralization of the neural tube, with ventral markers expanding throughout the entire ventral–dorsal axis and the absence of dorsal cell markers. These phenotypic characteristics closely resembled those in *Ptch1*-null mice [61]. Consistent with Sufu’s role in directly regulating Gli1, fibroblasts derived from *Sufu*-null embryos showed elevated Hh signaling activity, which could only be partially suppressed by PKA activation, but not by inhibiting Smo [73]. Through a series of genetic analyses with Sufu mutants and the three Gli mutant mice, Liu et al. [74] found that Sufu plays a positive role in the maximal activation of Hh signaling that defines the ventral-most cells’ fates. Specifically, neural progenitors at the floor plate (Foxa2-posotive) and for V3 interneuron (Nkx2.2-positive) failed to form in *Gli1*;*Sufu* double mutant embryos. This is potentially through Sufu’s protecting effect on Gli2 and Gli3, preventing these proteins from degradation [74].

To distinguish Sufu’s role during early stage of neural patterning and the later stage of cortical growth, Yabut et al. conducted an elegant study by generating two Sufu conditional-knockout mouse models [75]. The *Emx1-Cre*-mediated *Sufu* knockout started from E10.5 (Sufu-cKO-E10.5) mice, whereas the *hGFAP-Cre*-mediated Sufu knockout started from E13.5 (Sufu-cKO-E13.5) mice. These two mouse models revealed the changing role of Sufu in the developing telencephalon at different embryos stages. The mouse brain of Sufu-cKO-E10.5 displayed striking dorsal forebrain defects at the postnatal stages (P7), including a lack of olfactory bulbs, a thin and expanded cortex with entirely disrupted cortical laminations, expanded lateral ventricles, a disformed hippocampus, and an absence of corpus callosum [75]. In contrast, the forebrains of Sufu-cKO-E13.5 mice were grossly indistinguishable from those of control littermates at the postnatal stage [75].

To analyze cell proliferation and cortical lamination in Sufu-cKO-E10.5 mice, BrdU was introduced at various developmental stages (E12.5, E14.5, E16.5) and examined at P0. The results revealed that BrdU introduced at early stage (E12.5) displayed no significant difference compared to WT littermates. However, at later stages (E14.5 and E16.5), reduced BrdU labeling was found, and this coincided with a decrease in the specification of upper layer projection neurons. Consistently, the population of Tbr2-positive INPs progressively decreased by E16.5. This result suggests that Sufu is important for the maintenance of the apical progenitor pool. Further analysis also revealed various cell fate specification defects in Sufu-cKO-E10.5 mice. For instance, neurons labeled with BrdU at later stages were found to express the deep-layer marker Ctip2, instead of the expected marker for upper-layer neurons Cux1. Most strikingly, in the Sufu-cKO-E10.5 mouse brain, the ventral cell markers, Dlx2 and Mash1, were found to be expressed in a significant number of cells in the dorsal cortex at E14.5. This ventralization phenotype is likely attributable to the elevated level of Hh signaling in the dorsal forebrain, which may have potentially converted dorsal progenitor cell identities into ventral progenitor cell fates. In mice, GABAergic interneurons are exclusively generated from neural progenitors in the ventral cortex (Figure 2). However, recent studies suggest that in the human brain, RGCs in the dorsal cortex may also possess the capacity to generate GABAergic interneurons [76,77,78,79,80]. Since elevated Hh signaling is able to induce ventral progenitor cell fate in the dorsal cortex, it is intriguing to speculate that elevated Hh signaling could be the key factor enabling the generation of GABAergic interneurons in the dorsal cortex of the human brain and other gyrified species.

While early brain development appeared relatively normal in Sufu-cKO-E13.5 mice, at later embryonic stage (E16.5), the width of SVZ region was significantly expanded, together with a moderate expansion of VZ Width. This expansion correlated with an increase population of Tbr2-positive INPs, but not Pax6-positive RGCs [81]. Further, in the Sufu-cKO-E13.5 mouse brain, BrdU labeled higher density of dividing cells in SVZ, but fewer cells in VZ. These results suggest that loss of Sufu promotes transition of Pax6-positive RGCs into Tbr2-positive INPs. Consistently, at P7, there was an increased density in Cux1-positive upper layer neurons. Further analysis showed that loss of Sufu at this stage resulted in reduced levels of Gli3R, with no apparent changes in full-length Gli3 [81].

Collectively, these studies indicate that besides its role in influencing neural patterning during the early embryonic stages, the regulation of Hh signaling by Sufu after neural patterning is critical for both maintaining and specifying neural progenitors in the dorsal cortex. Moreover, this regulation is important for the specification of cortical neuronal cell fates, particularly for the later-born neurons that are destined for the upper cortical layers. These findings align with the hypothesis that elevated Hh signaling may serve a key role in the increased proliferation and expansion of neurons in gyrated brains.

### 3.5. The Transcription Factor, Gli, in Cortical Development

There are three Gli homologs in vertebrates. These three Gli transcription factors play both redundancy and reciprocal roles in Hh signal transduction, creating challenges to study individual Gli functions in cortical development. However, our understanding of the functions of these Gli proteins comes from research on neural patterning in various mouse Gli mutants. Therefore, we will provide a concise overview of the roles of Gli proteins in neural patterning and summarize the comparatively limited studies on the functions of Gli proteins in cortical neurogenesis.

The global knockout of *Gli1* in mice yields no discernible phenotypes [82,83], implying that Gli1 could be dispensable for neural development. This aligns with its recognized function as an amplifier of Hh signaling. However, earlier studies of overexpressing Gli1-induced phenotypes resembling excessive activation of Hh signaling [84,85].

Both Gli2 and Gli3 proteins are susceptible to proteolytic processing, leading to the generation of shorter forms that exhibit repressive functions in Hh signaling. Full-length Gli proteins function as transcriptional activators. Studies on mutant mice indicate that, during nervous system development, Gli2 primarily serves as an activator of the Hh pathway, while Gli3 predominantly functions as a repressor, although it may also exhibit activating properties in certain developmental events [86,87]. The repressive role of Gli3 was initially identified in an early study on a spontaneous mouse mutant, *extra-toes (Xt)*, characterized by a loss of function mutation in Gli3 [84]. The *Xt* mutant mouse exhibited polydactyly and central nervous system malformations, such as midbrain exencephaly, resembling the phenotypes associated with elevated Hh signaling [88,89,90]. Further studies revealed disrupted dorsoventral patterning of the telencephalon in the *Xt* homozygous mutant. The cell types of dorsal cortex are lost, and ventral-cell-type markers are ectopically expressed in the dorsal brain [90]. Furthermore, the mutant mice failed to develop the hippocampus and the choroid plexus in the lateral ventricle, due to defects in the invagination of the medial wall of the telencephalon [89,90].

Gli3’s repressor role in Hh signaling was demonstrated in *Shh^−/−^*;*Gli3^+/−^* double mutant. In this mouse model, the telencephalon defects in *Shh^−/−^* mutants are partially rescued by removing one copy of Gli3 [52]. The proboscis structure was replaced with larger telencephalon with two vesicles, and the missed ventral cell marker (Nkx2.1) was restored in *Shh^−/−^*;*Gli3^+/−^* mutants. The ventral (Gsh2) and dorsal (pax6) boundaries were also restored [52]. Most double homozygous mutants *Shh^−/−^*;*Gli3^−/−^* displayed exencephaly, a typical phenotype of elevated Hh signaling [5,82,91]. These results suggest the antagonist effect of Gli3 to Shh signaling. Interestingly, an activator role of Gli3 was found in the induction of the ventral most spinal cord progenitors. By expressing *Gli3* in *Gli2* locus, Bai et al. found that Gli3 has weak activator function by inducing the floor plate cell identity and V3 interneurons. This activator function of Gli3 is dependent on Gli1 transcription [86]. These results suggest the intricate complementary and antagonizing effects of the three Gli proteins.

To bypass neural patterning defects, Wang et al. generated *Gli3* condition-knockout mice after the neural patterning is complete [92]. In this mouse model, *Nestin-Cre* mediated removal of Gli3 gene in neural progenitors. In this *Gli3 cKO* mice model, deep-layer neuron production is prolonged at the expense of upper-layer neurons. Specifically, Cux1-positive upper-layer cortical neurons are missing in *Gli3 cKO* mice. BrdU incorporation suggests an early increase in RGC proliferation (at E14.5) but this reduced later, suggesting Gli3’s role in maintaining the RGC pool. Furthermore, the reduced RGC pool resulted in the decreased production of INPs. However, INPs exit the cell cycle prematurely in the *Gli3 cKO* brain, and this further prematurely depleted the INP pool. These results suggest that Gli3 plays important roles in maintaining the pool of RGCs and INPs via regulating the cell cycle dynamics of the progenitor cells [92].

Gli2 serves as the initial transducer of Hh signaling in the developing telencephalon, as it is present during initial Shh expression. *Gli2*-null embryos are embryonic lethal with brain malformations, such as a variably penetrant of exencephaly [93]. In the mildly affected brains that are not exencephalic, the cortices are much thinner, particularly in the proliferative zone VZ and SVZ. BrdU assay revealed reduced proliferation in VZ/SVZ at mid–late stages of cortical development. In the Gli2-null neocortex, Gli1 transcripts were absent, suggesting that Hh signaling is not activated.

Removing one copy of *Gli2* from *Gli1*-null background (*Gli1^−/−^; Gli2^−/+^*) leads to more severe defects such as the partial loss of the floor plate, reduced notochord, and loss of the pituitary and ventral diencephalon [83]. Interestingly, *Gli1/Gli2* double homozygous mutants (*Gli1^−/−^*;*Gli2^−/−^*) display less severe developmental defects in many tissues than Shh mutants, suggesting that without Gli1 and Gli2, Gli3 may partially compensate for some of roles as a transcription activator [86]. *Gli2* knockout was able to partially rescue the phenotypes associated with the elevated Hh signaling observed in *Ptch1-knockout* mice [82]. In *Gli2/Ptch1* double-knockout mice (*Ptch1^−/−^*;*Gli2^−/−^*), the ectopic Shh expression in the floorplate was rescued, so was the expanded expression of ventral markers (such as Nkx2.2). The rescue was prominent in the caudal neural tube, but not the brain or rostral neural tube, suggesting that Gli2 plays regional distinct roles [94]. Intriguingly, *Gli1* knockout failed to rescue these phenotypes.

In summary, mouse genetic studies revealed the complexity in the Gli protein network, which allows for the precise spatiotemporal regulation of Hh signaling. While much of the redundancy and complementary interactions among the three Gli proteins are elucidated through their involvement in the regulation of neural patterning, our understanding of their roles in the regulation of cortical neurogenesis remains limited. Another intriguing avenue for future research is related to the hypothesis of elevated Hh signaling in the formation of gyrencephalic brains. What evolutionary features the Gli protein network has acquired to facilitate the expansion of the neuronal production in the gyrified brain? Comparing Gli-target genes across different species has the potential to offer valuable perspectives.

### 3.6. Other Hh Signaling Regulators in Cortical Development

While we summarized studies on the core components of the Hh pathway during brain development, numerous studies have reported that other Hh modulators act to refine Hh signaling during brain development. One such important regulator is PKA, a potent Hh signaling suppressor that regulates the pathway by controlling Gli activation [95]. Genetic knockout of both isoforms of the PKA catalytic submit (*Cα* and *Cβ*) results in full activation of Hh signaling, leading to an open and ventralized neural tube resembling the phenotypes of *Ptch1* knockout [4]. Specifically, PKA activity at the centrosome and cilium directly participate in the regulation of Hh signaling [4,96]. Along this line, factors that impact intracellular cAMP levels at these two subcellular sites regulate PKA activity and Hh signaling. GPCR161 and Ankmy2 negatively regulate Hh signaling by controlling cAMP levels in the cilium, and their loss of function leads to defects in neural tube development [97,98]. Semaphorin–Neuropilin1 signaling contributes to the regulation of the overall intracellular cAMP levels to impact Hh signaling dynamics [99]. Additionally, the local pool of PDE4D at the centrosome modulates Hh signaling activity by regulating local cAMP levels [100].

During brain development, Shh and ligands for many other signaling pathways, such as FGF, Wnt, BMPs, and Retinoic Acid (RA), are found to be present in the cerebrospinal fluid (CSF) [101,102,103,104,105,106,107]. These signaling cue molecules could be detected by receptors located in primary cilia, which extend from the surface of the lateral ventricle and are submerged in the CSF [108]. Furthermore, over the course of cortical development, there is evidence that the protein components in the CSF exhibit some degree of variation [109]. Such variations could potentially influence the environmental cues received by RGCs, and subsequently impact their division mode, ultimately affecting the fate determination of their progeny. Indeed, patients with ciliopathy exhibit a variety of brain developmental deficits [110]. We will not summarize the roles of the primary cilium in brain development in this review. But it is important to note that mutations in genes related to cilium structure and ciliary protein trafficking machinery, including intraflagellar transport (IFT) and BBsome, are associated with a spectrum of cortical development defects [111,112]. For studies related to cilium and ciliary trafficking in brain development, we refer the readers to other more comprehensive reviews on this topic [110,113,114,115].

**Table 1 cells-13-00021-t001:** Mouse models and phenotypes in the Hh signaling pathway.

Mutation	Onset Stage of Knockout	Hh Signaling	Phenotype	Lethality Stage	Primary Publication
*Shh^−/−^*	Global	Down	Cyclopia, probiscus-like facial features, abnormal organ formation, reduced body size	Perinatal	[51]
*Shh^lox/lox^*;*Nestin^Cre^*	E12.5	Down	Decrease in MGE size, defective early oligodendrogenesis, fewer proliferating neural stem cells (NSCs) in postnatal SVZ and hippocampus	N/A	[72]
*Shh^lox/lox^*;*Emx1^Cre^*	E10.5	Down	Smaller dorsal telencephalon, abnormal neuron position and NSC characteristics	N/A	[53]
*Ptch^−/−^*	Global	Up	Open and overgrown neural tube	E9-10.5	[61]
*Ptch^+/−^*	Global	Up	Partially open neural tube, hindbrain defects, overgrown cerebellum, larger body size	N/A	[61]
*Ptch1^lox/lox^*;*Nestin^Cre^*	E12.5	Up	Surface of neocortex is folded with varying thickness, distorted brain structures in neocortex, MGE, hippocampus, medial cortex	E15.5	[62]
*Smo^−/−^*	Global	Down	Heart defect, cyclopia, loss of Left/Right symmetry	E9.5	[70]
*Smo^lox/lox^*;*FoxG1^Cre^*	E9	Down	Dorsalization of neural tube, loss of interneurons and oligodendrocytes	Perinatal	[71]
*Smo^lox/lox^*;*Emx1^Cre^*	E10.5	Down	Smaller telencephalon, defects in neuronal migration, significant loss of oligodendrocytes	N/A	[53]
*Smo^lox/lox^*;*GFAP^Cre^*	E13.5	Down	Small brain, fewer INPs, fewer bRGCs	N/A	[44]
*SmoM2* (drive by *GFAP^Cre^*)	E13.5	Up	Folding of the cingulate cortex, higher number of bRGs and INPs	N/A	[44]
*Sufu^−/−^*	Global	Up	Open ventralized neural tube	E9.5	[73]
*Sufu^−/+^*	Global	Up	Normal growth, fertile, develop Gorlin-like features	N/A	[73]
*Sufu^lox/lox^*;*GFAP^Cre^*	E13.5	Up	No obvious phenotype, survive into adulthood, expanded VZ and SVZ	N/A	[75]
*Sufu^lox/lox^*;*Emx1^Cre^*	E10.5	Up	Large cortical surface, no olfactory bulb, expanded lateral ventricles, thinner cortical layer	Death before weaning	[75]
*Gli1^−/−^*	Global	N/A	Normal, viable	N/A	[83]
*Gli2^−/−^*	Global	Down	Small lungs and fused lobes, no notochord regression, loss of pituitary, craniofacial defects	Perinatal	[83]
*Gli2P^1−4^* (S/A mutation at four PKA sites)	Knock-in	Up	Exencephaly, partially open neural tube, extra anterior digit, enlarged facio-cranial features	Between E14.5 to birth	[5]
*Xt* (Spontaneous loss-of-function mutation in *Gli3*)	Global	Up	Dorsalized neural tube, reduced cortical size, absent hippocampus and choroid plexus	N/A	[88]
*Gli3^lox/lox^*;*Nestin^Cre^*	E12.5	Up	Loss of upper layer projection neurons (PNs), defects in cortical neuron specification and positioning	N/A	[92]
*Shh^−/−^*;*Gli3^−/−^*	Global knockout	N/A	91% exencephalic, relatively normal telencephalon, missing dorsal midline structure, normal pan-ventral genes such as Dlx2 Gsh2, Nkx2.1	N/A	[52]
*Shh^−/−^*;*Gli3^+/−^*	Global	Down	Partial rescue of Shh-null phenotype: two discernable eyes, reduced probiscus, partial dorsal–ventral patterning rescue	N/A	[52]
*Gli1^−/−^*;*Gli3^+/−^*	Global	Up	Viable, polydactylyl	N/A	[83]
*Gli1^−/−^*;*Gli2^−/−^*	Global	Down	Decreased viability by E18.5, loss of pituitary tissue, lung lobes defects	Perinatal	[83]
*Gli1^−/−^*;*Gli2^+/−^*	Global	Down	Some loss of ventral spinal cord, small lungs	Perinatal	[83]

## 4. Summary and Future Perspective

Studies in various mouse genetic mutants revealed the complex interplay among Hh pathway components and the modulation of Hh signaling by other regulators. By delineating the phenotypes of single or combined genetic mouse models, we have gained invaluable insights into the critical roles of Hh signaling in the regulation of brain development, including neural patterning, and the late developmental events in the neocortex. Mutations in Hh pathway components are linked to a spectrum of human brain developmental disorders. In the past, numerous pharmacological modulators of Hh signaling have been developed, mostly targeting the Hh-related tumors [116]. These chemical modulators also offer promise for the treatment of neurological disorders associated with Hh signaling defects.

In the rodent brain, after neural patterning, Shh exhibits high expression in the ventral cortex, playing a crucial role in the specification and production of GABAergic interneurons. However, in the human neocortex, neural progenitors have been observed to possess the capacity to generate GABAergic neurons [77,78,79,80]. This phenomenon is potentially linked to the extension of Hh signaling into the dorsal cortex. Further, the elevated Hh signaling in the neocortex is also suggested to correlate with the significantly increased neuronal production in the gyrified brain. This gives rise to intriguing questions: What specific step in the signaling transduction sees the increase in Hh signaling in the human neocortex? What evolutionary features contribute to the expansion in Hh signaling in the gyrified brain? Insights into these questions could be obtained through studies with in vitro models of human cortical development, such as human cerebral organoid. Additionally, since Hh signaling regulates the expression of sets of target genes, cross-species comparisons of the differential expression of these Hh target genes could also yield valuable insights.

## Figures and Tables

**Figure 1 cells-13-00021-f001:**
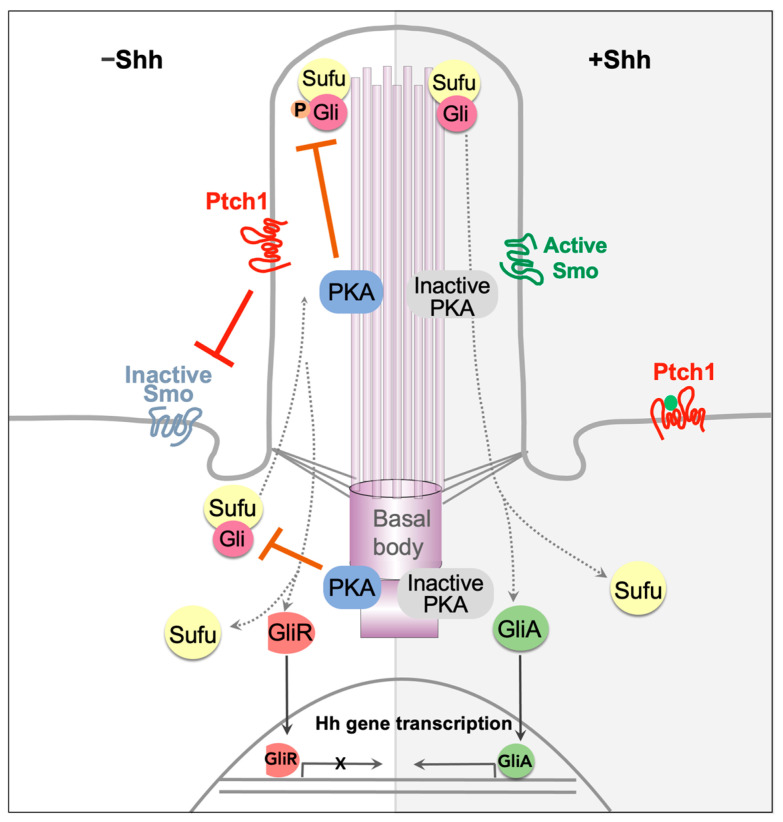
Transduction of Hedgehog signaling. In the absence of the ligand, Shh, Ptch1 in the cilium inhibits Smo activity. Sufu sequesters the Gli transcription factors, and the complex translocates through the primary cilium. Gli is phosphorylated by PKA at the basal body and cilia, priming it for proteolytic processing into repressor form. As a result, the transcription of Hh target genes is off (**left**). Upon Shh binding to Ptch1, the complex exits the cilium. Subsequently, Smo enters the cilium and inactivates PKA. This results in Gli dissociation from the Sufu and its ultimate activation. Active Gli enters the nucleus to initiate the transcription of Hh target genes (**right**).

**Figure 2 cells-13-00021-f002:**
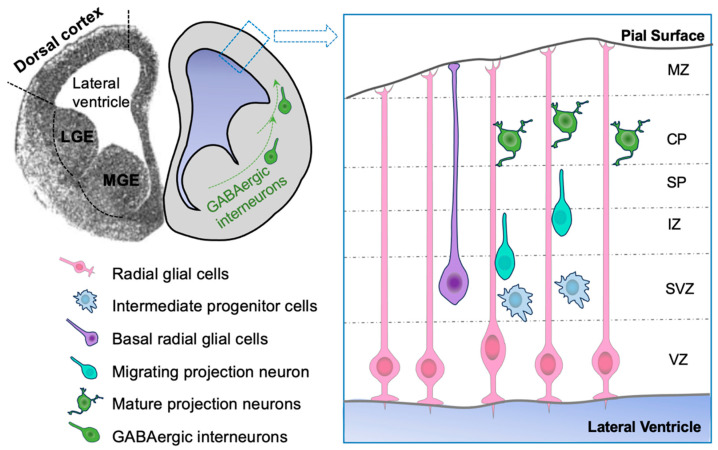
The organization and cell types in the developing neocortex (adapted from Toresson et al. 1999). The ventricular zone (VZ) contains the cell body of radial glial cells (RGCs) that attach to the pia surface via the basal process, and to the lateral ventricle surface via the apical process. RGCs may directly generate neurons via asymmetric cell division, and the differentiated neurons migrate to the cortical plate along the RGC’s radial fibers. RGCs also generate multipolar intermediate neural progenitors (INPs) that divide in the subventricular zone (SVZ) to produce neurons. Basal radial glial cells (bRGCs) have a basal process attached to the pial surface, but no apical processes. bRGCs are more abundant in gyrified brains. In rodents, all GABAergic interneurons are generated in the ventral brain, and migrate tangentially to the dorsal cortex; glutamatergic projection neurons are generated in the dorsal brain and migrate radially to the cortical plate. VZ: ventricular zone; SVZ: subventricular zone; IZ: intermediate zone; SP: subplate; CP: cortical plate; MZ: marginal zone. LGE: lateral ganglionic eminence; MGE: medial ganglionic eminence.

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
