# Peer review of "Hedgehog Signaling in Cortical Development"

_cells, 2023, doi:10.3390/cells13010021_

Round 1
Reviewer 1 Report
Comments and Suggestions for Authors
The authors of the paper “Hedgehog signaling in cortical development” seek to review the role of Hedgehog (Hh) signaling in the development of the cortex of the mammalian brain with the focus on neural progenitor proliferation and neuron production. For this purpose they summarize the results from genetic mouse models targeting the ligand Shh, the receptor Ptch1, the GPCR-like transducer Smo, the intracellular transducer Sufu, and the three Gli transcription factors.
The review is in general well written. Exceptions are the use of abbreviations (please see e.g. INPs in line 65; missing explanation of abbreviations in Figure legend 2 e.g. LGE and MGE or missing explanation of all the abbreviations used in Table 1) or misspellings (such as “PTCH1 duplicating” in line 208; or “the cortical appeared….” in line 220; “Gli1 is not indispensable” in lane 385 that better should read “Gli is dispensable). The review is also easy to understand to the reader.
However, I was missing several things:
1) The authors simply summarize the findings in mouse models, in which Hh signaling components have been modulated in the brain (sections 3.1 to 3.5). In several sections (e.g. 3.2. and 3.5.) I was missing what the respective studies exactly tell us about the function of Hh signaling in cortical development: thus, the studies are frequently about general brain development and therefore, in my opinion, the title of the review is not adequate.
2) I was missing the information about the exact expression pattern of the Hh signaling components in the cortex and its precursor cells that would have helped to better understand the role of Hh signaling in cortex development.
3) Similarly, I was missing a column in Table1 that summarizes the conclusions concerning the role of Hh signaling in cortex development. In addition, a drawing where and how the different Hh signaling components act during cortex development would be helpful.
4) I also think that the authors should point to the weakness of mouse studies when analyzing Hh signaling in cortex development and should discuss in more depth the differences between the human (with gyry) and rodent (without gyry) cortex.
Please correct misspellings such as “PTCH1 duplicating” in line 208; or “the cortical appeared….” in line 220; “Gli1 is not indispensable” in lane 385 that better should read “Gli is dispensable.
Reviewer 2 Report
Comments and Suggestions for Authors
This review concentrates on canonical Shh signaling in mouse cortical development. It mainly discusses the canonical Shh signalling pathway and I think this should be mentioned where appropriate (even if most readers would know this anyway). It would be interesting to compare with human brain development, perhaps in a separate short segment as this manuscript is not about human brain development. More consideration about other factors that modify or interact with Hh signalling might be useful when explaining the findings in different genetic mouse mutants. A list of abbreviations would be helpful. Not all abbreviations have been listed in some of the figure legends. There are some grammatical errors, although these generally do not interfere with understanding of the manuscript. There also needs to be attention to gene and protein symbols and whether the findings relate to Shh or Hh in general.
Comments on the Quality of English LanguageMinor editing of English language required
Round 2
Reviewer 1 Report
Comments and Suggestions for Authors
The authors have addressed all my questions.